# OpenReview forum: "On Robustness and Chain-of-Thought Consistency of RL-Finetuned VLMs"
_ICLR.cc/2026/Conference — Submitted to ICLR 2026_

### Official Review · Reviewer_j4PE · 2025-10-28

**Soundness:** 2
**Presentation:** 1
**Contribution:** 2
**Rating:** 2
**Confidence:** 4

**Summary:**

This paper investigates the robustness and chain-of-thought (CoT) consistency of reinforcement-finetuned vision-language models (VLMs). The authors introduce controlled textual perturbations (e.g., Wrong-Caption and Wrong-Think) to test whether these models can resist misleading text and remain visually grounded. Results show that although RL fine-tuning improves benchmark accuracy, it reduces reasoning faithfulness and heightens sensitivity to misleading context. To mitigate these effects, the authors propose data augmentation and faithfulness-based reward methods that enhance both robustness and reasoning coherence.

**Strengths:**

The paper focuses on an important question—how Reinforcement Finetuning affects the reasoning faithfulness of large vision-language models.

**Weaknesses:**

1. The overall contribution of this paper is limited. It lacks technical innovation, does not present a solid benchmark, and offers no particularly insightful experimental findings.

    a. The paper primarily analyzes existing RL-trained VLMs through text perturbations. However, such perturbation-based evaluation has been extensively explored in prior works, such as [1].

    b. The benchmark proposed in this paper is mostly an extension of existing datasets with additional annotations of incorrect text or captions. This extension is incremental and does not provide substantial novelty or methodological advancement. Besides, it is not clear whether the additional annotations will be released.

    c. The proposed data augmentation strategy resembles approaches introduced in previous studies, such as [1]. Furthermore, the faithfulness-based reward function is conceptually similar to previously explored *process reward models* used in prior work, such as [2]

[1] Chen et al. PerturboLLaVA: Reducing Multimodal Hallucination with Perturbative Visual Training.

[2] Zhang et al. The Lessons of Developing Process Reward Models in Mathematical Reasoning.


2. The results in Table 1 show only marginal improvements or even degradations. Moreover, the evaluation relies on a single prompt configuration, making it difficult to assess the robustness and reliability of the reported results. To improve experimental validity, multiple prompt settings should be tested to evaluate consistency across prompt variations.


3. The assessment of faithfulness and reasoning trace quality depends solely on a single large-model judge. The absence of human evaluation or inter-annotator validation undermines the reliability of the reported metrics. It remains unclear whether the automated judgments accurately reflect true reasoning quality.

4. The used datasets assess simple spatial patterns and do not reflect complex multimodal reasoning or real-world understanding. It is unclear whether improvements or degradations on such tasks can generalize to more complex real world visual understanding.

5. The implementation details of experiments in Section 3 are unclear. Though the author provide details on how to generate captions and initial think, it is unclear how these annotations are used in training.

**Questions:**

1. How does your method differ from previous perturbation-based evaluations and previous design on process reward (e.g., [1], [2])?
2. Could you describe more clearly how the “initial think” and caption data are used in training?
3. Are the findings consistent when tested on more complex used, multimodal reasoning datasets, such as MMBench, V* Bench, MME, GQA?

---

> ### Author Response · Authors · 2025-11-21
>
> We thank the reviewer for their thorough review of our work. We’re glad they believe understanding how reinforcement fine-tuning affects the reasoning faithfulness of VLMs is an important problem. We address their concerns below.
>
> ### Weaknesses
>
> > a. The paper primarily analyzes existing RL-trained VLMs through text perturbations. However, such perturbation-based evaluation has been extensively explored in prior works, such as [1].
>
> Thank you for pointing out this connection; we now cite and explicitly compare to PerturboLLaVA in Related Work. While PerturboLLaVA targets dense captioning and trains models to be less hallucination-prone via adversarial text/visual perturbations (using the unperturbed trace as the label), our focus is on multiple-choice visual reasoning. Our perturbations are procedurally generated so that misleading text explicitly supports a wrong option while the image is fixed, allowing us to probe cross-modal conflict and CoT faithfulness. In addition, we specifically study the RL fine-tuning regime by both analyzing existing RL-finetuned VLMs and running new RL experiments to track how robustness and CoT consistency evolve over training.
>
> > b. The benchmark proposed in this paper is mostly an extension of existing datasets with additional annotations of incorrect text or captions. This extension is incremental... Besides, it is not clear whether the additional annotations will be released.
>
> Our goal is to provide a standardized, controlled perturbation suite for visual reasoning VLMs. We unify multiple benchmarks under a common perturbation protocol designed to test robustness to modality conflict: procedurally generated text perturbations systematically bias models toward a specific wrong answer, and we measure how often they follow this suggestion despite unchanged visual evidence. The same perturbation types are used during RL finetuning to study how training choices affect both robustness and CoT consistency over time, revealing the accuracy–faithfulness trade-off and “faithfulness drift.” We will publicly release all perturbation scripts and annotations so others can reproduce and extend these stress tests. We believe the combination of (i) carefully constructed conflicting text, (ii) faithfulness-oriented evaluation, and (iii) RL-dynamics analysis offers value beyond prior perturbation work focused on caption quality or generic hallucination.
>
> > c. The proposed data augmentation strategy resembles approaches introduced in previous studies, such as [1]. Furthermore, the faithfulness-based reward function is conceptually similar to... prior work, such as [2]
>
> We appreciate the connections to PerturboLLaVA [1] and process reward models [2], but our setting differs in both mechanism and objective. Rather than training a separate reward model, we use an off-the-shelf LLM-as-judge on the full chain of thought and final answer, yielding a lightweight faithfulness-aware reward without an additional PRM training stage. While [2] designs PRMs for mathematical reasoning, we ask whether augmentation plus a simple faithfulness reward suffices to make VLMs robust when text and image conflict. Empirically, it does not: under faithfulness constraints, the augmented data induces shortcut strategies where the model learns to “trust” right-caption/right-think patterns (which are frequently rewarded) instead of using CoT to resolve cross-modal conflicts. This suggests that stronger interventions (e.g., vision-based rewards or traces that explicitly correct misleading text) are needed. This also contrasts with PerturboLLaVA, which uses unperturbed chains of thought as ground truth to train models to ignore perturbations; our aim is for models to explicitly reason about them, not simply discard them, since textual context can be privileged rather than always corrupted.
>
> > 2. The results in Table 1 show only marginal improvements or even degradations. To improve experimental validity, multiple prompt settings should be tested to evaluate consistency across prompt variations.
>
> The intent of Table 1 is precisely to show that switching from default prompting to “Stop-Think” yields a mix of gains and degradations, with no systematic pattern across models and tasks. Our goal is not to advocate for Stop-Think as a universally better prompting strategy, but to highlight that, unlike in text-only settings where “No-Thinking” shows clearer trends [1], multimodal reasoning under CoT suppression is highly heterogeneous. We also analyze an additional prompt configuration in Appendix D.2, allowing models to abstain; we find performance generally decreases when abstention is enabled, especially under Wrong-Caption/Wrong-Think. If there are specific alternative prompt formats the reviewer considers important, we would be happy to evaluate those as well.
>
> [1] Ma, Wenjie, et al. "Reasoning models can be effective without thinking." arXiv preprint arXiv:2504.09858 (2025).

---

> > ### Author Response · Authors · 2025-11-21
> >
> > > 3. The assessment of faithfulness and reasoning trace quality depends solely on a single large-model judge.
> >
> > We agree that having additional evaluators would strengthen the claims and thus we have added faithfulness results across two more judge models, GPT-OSS-120B and Llama-3.1-70B-Instruct. We chose these models to span different model families and training data distributions. We also compute inter-annotator agreement metrics across the three models. Please see the general comment and updated manuscript for details; overall, we observe that the results are consistent, with high inter-annotator agreement across judge models.
> >
> > > 4. The used datasets assess simple spatial patterns and do not reflect complex multimodal reasoning or real-world understanding. It is unclear whether improvements or degradations on such tasks can generalize to more complex real world visual understanding.
> >
> > Thank you for raising this point. Our choice to focus on “simple” spatial tasks (counting, identity, 2D/3D spatial relations) was motivated by being foundational prerequisites for more complex multimodal reasoning: if a model can be systematically pushed off the correct answer distribution on basic, highly controlled visual tasks by simple prompt injections, then it is hard to trust its behavior on more open-ended, real-world queries.
> >
> > We also recognize your broader concern about generalizing beyond the “basic spatial” regime to richer real-world understanding. In response to this, we have added three additional benchmarks—MMBench, MME-RealWorld-Lite, and V*-Bench—which place more emphasis on general VQA and diverse, real-world semantics rather than only spatial layouts. In the current revision, we include these new datasets in our analysis of open-source VLMs and CoT faithfulness in Appendix D.1, showing that the kinds of robustness and grounding issues we highlight do appear beyond narrowly structured spatial tasks. We are actively running the same experiments on our data-augmented checkpoints for these benchmarks; if having full results on these other datasets is crucial for the reviewer’s overall assessment, we would very much appreciate an explicit indication so we can prioritize these experiments accordingly within our compute and space budget.
> >
> > > 5. The implementation details of experiments in Section 3 are unclear. Though the author provide details on how to generate captions and initial think, it is unclear how these annotations are used in training.
> >
> > We agree that the description of how the think and caption augmentations are incorporated during RL finetuning could be made clearer. In Section 3, our intention was to convey that these augmented annotations are used as conditioning inputs during rollout generation, not as separate supervision signals. Specifically, each question in the SAT2 or Pixmo Count datasets has four possible augmentations: a wrong or correct initial thinking string, and a wrong or correct caption. During finetuning, one of these augmentations is randomly applied with a 10% probability (for a total of 40% augmented rollouts), while the remaining 60% of samples are presented in their original form. This information is provided in Section 3.1.
> >
> > The motivation for including both correct and incorrect captions (and likewise for thinking strings) is to prevent the model from overfitting to a trivial heuristic. If only wrong captions were used, the model would quickly learn to always “invert” or distrust the provided context—essentially memorizing that every caption is misleading—rather than learning to evaluate the reliability of auxiliary information. By mixing both correct and incorrect augmentations, the model instead learns to discern whether the contextual signal (caption or think string) aligns with the visual evidence and question, better reflecting realistic inference conditions. We appreciate the reviewer highlighting this point and have revised the manuscript in Section 3.1. to more clearly communicate our setup.
> >
> > ### Questions
> >
> > > How does your method differ from previous perturbation-based evaluations and previous design on process reward (e.g., [1], [2])?
> >
> > Please see our response to your specific points in``Weaknesses’’ above and let us know if we can provide any further clarification.
> >
> > > Could you describe more clearly how the “initial think” and caption data are used in training?
> >
> > Please see ``Weaknesses’’ above and let us know if we can provide any further clarification.
> >
> > > Are the findings consistent when tested on more complex used, multimodal reasoning datasets, such as MMBench, V* Bench, MME, GQA?
> >
> > Please see ``Weaknesses’’ above for our discussion about additional datasets used for evaluation. We appreciate the reviewer’s specific feedback about additional datasets to try.
> >
> > Once again, we thank the reviewer for their constructive feedback and thoughtful suggestions, and we would be glad to provide any additional clarifications or experiments as needed.

---

> > > ### Comment · Reviewer_j4PE · 2025-11-22
> > >
> > > Thanks for the detailed clarification on the experimental setup and additional results. Several of my concerns are addressed (i.e., training details, results on more complex benchmarks, and evaluation faithfulness). I appreciate the authors’ efforts and accordingly raise my score to 4.
> > >
> > > However, my main concern regarding the contribution of the work remains.
> > >
> > > The overall training and evaluation paradigm is highly similar to a broad body of existing methods: noise or perturbations are added to the inputs, and the model is then trained on these noisy/perturbed samples to improve robustness. This framework has been extensively studied in prior work, beyond the specific references I initially provided. From my perspective, the paper primarily transfers established analysis techniques and training strategies to RL-trained VLMs, rather than proposing a fundamentally new methodology.
> > >
> > > Similarly, the proposed data extension procedure largely relies on prompting existing LLMs/VLMs to generate additional annotations. Given that leveraging large models to synthesize or augment data is now a standard practice, it is difficult to view this as a novel pipeline. The paper does not clearly articulate a substantial methodological advance beyond this widely adopted pattern.

---

> > > > ### Author Response · Authors · 2025-12-03
> > > >
> > > > We thank the reviewer for promptly taking the time to re-evaluate our work, as well as for acknowledging the additional experiments and evaluator checks that address several of the earlier concerns.
> > > >
> > > > Our intention was not just to reinstantiate existing pipelines, but to use these tools to probe a regime that, to our knowledge, has not been previously systematically analyzed with textual perturbations and adversarial-style augmentation: RL-finetuned VLMs with chain-of-thought. In this setting, we believe our analysis surfaces qualitatively new and practically important phenomena that go beyond standard robustness observations: (1) a consistent accuracy–faithfulness drift under RL, (2) brittleness to simple, procedurally generated cross-modal perturbations, and (3) the emergence of shortcut strategies even under a faithfulness-aware reward. We see these as new insights into how existing techniques behave in this specific, increasingly relevant deployment regime, rather than as a claim of proposing an entirely new learning algorithm.
> > > >
> > > > In the first part of our work, we deliberately use simple but controlled text perturbations that enforce a specific wrong alternative, and show that even methods which go beyond 0–1 correctness (e.g., adding auxiliary rewards for adding bounding boxes in the intermediate CoT) can still yield unfaithful reasoning and modality blindness, which we believe is a novel observation. In the second part of our work, the fact that a “standard” robustness recipe (noisy/perturbed inputs plus RL and data augmentation) still fails to produce visually grounded and CoT-consistent behavior in this setting is precisely the negative result we think is useful to surface. In similar research areas (e.g., RLHF/RLVR for text-only LLMs), much progress has come from carefully analyzing how classical RL and augmentation tools behave in a new regime; our work aims to play a similar role for RL-tuned VLMs. Similarly, although using large models for data augmentation is now common, what is distinctive here is how we structure the augmented data into these right-/wrong-caption and right-/wrong-think variants for a fixed image and answer, and use this to probe cross-modal conflict, robustness, and CoT faithfulness before and after RL across multiple benchmarks.
> > > >
> > > > In short, our goal is not to introduce a novel pipeline or chase SOTA, but to provide a careful empirical study showing that current RL finetuning paradigms for VLMs can yield models whose output distributions are highly susceptible to simple misleading text cues and whose chains of thought drift away from faithful visual grounding, even under faithfulness-aware objectives. Finally, at the suggestion of Reviewer 2c19, we have added an entropy-based analysis (Appendix D.1) to our manuscript. This analysis reinforces the utility of our perturbation framework by uncovering distinct behavioral profiles that top-line accuracy metrics miss. Specifically, we find a dichotomy in how models handle conflict: “stubborn experts” show high correlation between base-confidence and robustness but achieve this by ignoring the CoT, while “obedient reasoners” are faithful to the CoT but exhibit brittle confidence profiles. We believe this demonstrates the value of our methodology: it is not just a training recipe, but a diagnostic tool that reveals how different post-training setups can drive models toward fundamentally different trade-offs between robustness, calibration, and faithfulness. Please refer to our Official Comment for the full details of this update.

---

### Official Review · Reviewer_h5wX · 2025-10-30

**Soundness:** 3
**Presentation:** 3
**Contribution:** 2
**Rating:** 4
**Confidence:** 3

**Summary:**

This paper systematically evaluates the robustness and COT faithfulness of RL-tuned vision language models on basic visual reasoning, including counting, identity, and 2D and 3D spatial relations on static images. The authors augment established benchmarks: 3DSRBench, CV-Bench, Spatial-MM, and WhatsUp and add controlled textual perturbations that keep the image unchanged:: Stop-Think, Wrong-Think, and Wrong-Caption. Across five RL-finetuned models derived from Qwen2.5-VL-7B-Instruct, the study measures accuracy, robustness, and CoT consistency.

**Strengths:**

Focusing on robustness and faithful, visually grounded reasoning, the paper uses simple, controlled textual perturbations to effectively probe modality conflict.
By analyzing training dynamics, the paper indicates an accuracy–faithfulness tradeoff, shows that augmentation improves robustness while faithfulness continues to drift, and finds that adding faithfulness to the reward aligns CoT with answers yet becomes unstable when combined with augmentation, yielding limited robustness gains.

**Weaknesses:**

The paper mainly reveals the accuracy–faithfulness disconnect and sensitivity to textual perturbations, but does not provide training or inference method that can be readily reused.
The augmentation strategies with wrong-think and wrong-caption yield clear in-distribution improvements, but evidence for transfer across datasets and tasks is limited.
Out-of-distribution performance is under-reported, including results on different data sources and task types
Formatting error: “n Appendix D.1 we show that” seems it should be “In Appendix D.1 we show that.”

**Questions:**

Please report results on  spatial-relation datasets from different sources, outside the training mix, to quantify performance under distribution shift
Evaluate transfer without changing the training method (e.g., spatial relations, spatial relations → geometric reasoning) to distinguish answer accuracy from visually grounded reasoning.
Report out-of-distribution variants for Wrong-Think and Wrong-Caption separately.

---

> ### Author Response · Authors · 2025-11-21
>
> We thank the reviewer for their careful feedback and are glad that our focus on robust, visually grounded reasoning and the controlled perturbations came through clearly. We address their comments below.
>
> ### Weaknesses
>
> > The paper mainly reveals the accuracy–faithfulness disconnect and sensitivity to textual perturbations, but does not provide training or inference method that can be readily reused.
>
> We would first like to emphasize that our approach is designed to be reusable. We demonstrate this by applying the same perturbation and evaluation pipeline across a diverse set of benchmarks and training datasets: in Appendix B.2 we detail how we add textual perturbations for each selected benchmark, including concrete examples of Wrong-Caption and Wrong-Think perturbations, and in Appendix C we describe how we augment two distinct RL training datasets (SAT2 and Pixmo-Count) with correct and incorrect captions and thinking strings alongside Section 3.1. To support reproducibility, Appendix B.4 specifies exactly how we obtain faithfulness results from a collection of model responses, including the precise prompt used for the Qwen3 judge, and Appendix C provides RL finetuning details and prompts for data augmentation; combined with our use of open-source models, datasets, and the verl framework, we intend these details to be sufficient for others to replicate or adapt our pipeline. Given the consistency of the observed accuracy–faithfulness disconnect and sensitivity to textual perturbations across these varied settings, we expect the same methodology to transfer to similar benchmarks and datasets, and we plan to further facilitate reuse by releasing our full codebase (for evaluation and RL training) and the augmented datasets in the final version.
>
> > The augmentation strategies with wrong-think and wrong-caption yield clear in-distribution improvements, but evidence for transfer across datasets and tasks is limited. Out-of-distribution performance is under-reported, including results on different data sources and task types
>
> We appreciate this comment and are happy to clarify the data splits and the extent of distribution shift in our experiments. First, all evaluation benchmarks used in our work—CVBench, WhatsUp, SpatialMM (Obj/Multihop), or 3DSRBench—are not included in our RL training data (refer to Section 3.1). Our RL finetuning is performed on a mixture of SAT2 and Pixmo-Count (optionally with Geometry3K), and we selected this combination precisely to avoid any overlap in images or questions with the benchmarks on which we report robustness results. SAT2 in particular only contains synthetic images from distinct scenes, whereas the evaluation benchmarks are composed of real or more naturalistic images, so the gains from our Wrong-Think/Wrong-Caption augmentation already reflect transfer from synthetic images to separate datasets and data sources. The original SAT paper noted this transfer in the SFT setting, and we confirm this also holds for RL fine-tuning. We will mention making this choice and the non-overlap more explicit in the revised text.
>
> Second, we agree that it is valuable to probe transfer beyond the “basic spatial / counting” regime that our core benchmarks target. In response to this comment and to Reviewer j4PE’s related suggestion, we have added three additional evaluations: MMBench, MME-RealWorld-Lite, and V*-Bench, which emphasizes more general VQA and real-world understanding rather than primarily spatial relations. In the current revision, these new datasets are integrated into our first set of experiments on open-source VLMs and CoT faithfulness, extending our conclusions about robustness and faithfulness to a broader range of tasks and domains.
>
> We are actively running the full Wrong-Think/Wrong-Caption + RL augmentation experiments on our RL-finetuned checkpoints for these new benchmarks as well; we plan to include these OOD results in the next revision. If having full RL results on MMBench / MME-RealWorld-Lite / V*-Bench is a key factor for the reviewer’s final score, we would greatly appreciate their indicating this explicitly so that we can prioritize reporting those additional experiments accordingly.
>
> > Formatting error: “n Appendix D.1 we show that” seems it should be “In Appendix D.1 we show that.”
>
> Thank you for identifying this typo, we have made the edit in the draft.

---

> > ### Author Response · Authors · 2025-11-21
> >
> > ### Questions
> >
> > > Please report results on spatial-relation datasets from different sources, outside the training mix, to quantify performance under distribution shift
> >
> > Thank you for this suggestion, please see our response in ``Weaknesses’’ above. If the reviewer has specific additional datasets in mind that they consider especially important for demonstrating OOD generalization, we would be very grateful if they could name them; we will do our best to incorporate those evaluations subject to space and compute constraints.
> >
> > > Evaluate transfer without changing the training method (e.g., spatial relations, spatial relations → geometric reasoning) to distinguish answer accuracy from visually grounded reasoning. Report out-of-distribution variants for Wrong-Think and Wrong-Caption separately.
> >
> > With the new results on MMBench, MME-RealWorld-Lite, and V*-Bench, we hope this emphasizes more general VQA and real-world understanding rather than only spatial relations. Regarding the suggested “spatial relations → geometric reasoning” transfer: we agree that studying transfer from visually grounded spatial tasks to more abstract mathematical or geometric reasoning (e.g., text-only geometry or multi-step symbolic math) is a compelling direction. However, we see this as beyond the scope of the current work because  the intent of the paper is precisely to focus on these basic visual reasoning tasks (counting, identity, 2D/3D spatial relations) because we see them as prerequisite skills for harder visual reasoning: the benchmarks are intuitively “easy” and structurally controlled, yet we find that the model’s answer distribution can still be strongly and systematically influenced by simple, procedurally generated prompt injections (Wrong-Think/Wrong-Caption), which makes them an especially clean testbed for studying visually grounded robustness and CoT faithfulness. Moving to non-visual or heavily symbolic mathematical reasoning would introduce additional confounds like the model’s mathematical capabilities.
> >
> > Once again, we thank the reviewer for their constructive feedback and thoughtful suggestions, and we would be very happy to provide further clarifications or additional experiments—especially on specific datasets or transfer settings the reviewer deems most informative—as we continue to refine this work.

---

### Official Review · Reviewer_2c19 · 2025-10-31

**Soundness:** 3
**Presentation:** 4
**Contribution:** 3
**Rating:** 6
**Confidence:** 3

**Summary:**

This paper presents a systematic empirical study of the robustness and chain-of-thought (CoT) consistency of RL-finetuned vision-language models (VLMs). By introducing controlled textual perturbations—misleading captions and reasoning prefixes—the authors reveal that RL-tuned models are often misled by textual cues and exhibit an accuracy–faithfulness trade-off. They further analyze the effects of adversarial augmentation and faithfulness-aware rewards. The findings provide valuable diagnostic insights into how RL fine-tuning affects multimodal reasoning reliability.

**Strengths:**

+ Proposes a clear and reproducible textual perturbation framework to probe VLM robustness.

+ Identifies a consistent accuracy–faithfulness trade-off during RL fine-tuning.

+ Covers a wide range of recent RL-based multimodal reasoning models and benchmarks.

+ Analysis is careful, and the findings are both timely and practically relevant.

**Weaknesses:**

+ The paper remains primarily empirical, without a formal theoretical explanation or principled model-level intervention to mitigate the observed trade-off.

+ The faithfulness-as-reward experiments, though conceptually interesting, are underexplored; their instability and optimization dynamics merit deeper quantitative analysis.

+ The study relies on a single large-language-model judge (Qwen3-32B) to assess reasoning faithfulness, which may introduce evaluation bias; cross-validation with other judgment models would strengthen the claims.

+ While the work discusses potential solutions (e.g., richer reward signals, uncertainty modeling), these remain qualitative suggestions rather than systematically validated interventions.

**Questions:**

+ Have the authors considered developing a formal metric or theoretical framing for “faithfulness drift”? For instance, could the trade-off between accuracy and CoT consistency be modeled via reward attribution entropy or causal influence metrics?

+ Can the authors provide quantitative evidence—such as variance across seeds, reward gradients, or convergence plots—to substantiate the claim of unstable training dynamics?

+ Have alternative evaluators been tested to confirm that the faithfulness judgments are not artifacts of Qwen3’s inductive biases?

+ Could the authors empirically evaluate one or more of the suggested remedies (e.g., uncertainty-aware reward, contrastive consistency loss) to demonstrate their potential effectiveness?

---

> ### Author Response · Authors · 2025-11-21
>
> We thank the reviewer for taking the time to review our work and are especially encouraged that they found the textual-perturbation framework clear and reproducible, the analysis well-presented, and the findings timely. We’re also glad that the consistency of the accuracy–faithfulness trade-off and our coverage of recent RL-based VLMs and benchmarks resonated, and we address their comments in detail below.
>
> ### Weaknesses
>
> > The paper remains primarily empirical, without a formal theoretical explanation or principled model-level intervention to mitigate the observed trade-off.
>
> We agree that a formal theoretical explanation and more principled model-level interventions would be an important next step. That said, we believe our negative results point to where existing intuitions about chain-of-thought may break down in visual domains. In particular, in the visual domains we study, it may be the case that chain-of-thought does not automatically act as a faithful “window” into the model’s underlying computation. Under RL, the CoT becomes an additional degree of freedom in the output distribution: the model can freely reshape its textual reasoning to optimize the reward, while the real decision-making and feature transformations may be happening in latent representations (we now explicitly discuss this in the Conclusion). As a first principled intervention, we do introduce targeted text perturbations together with a faithfulness-aware reward, but our experiments show that this combination can still induce shortcut policies that match the reward signal without genuinely grounding the reasoning in the image. This suggests that faithfulness is not a free byproduct of CoT, but an additional constraint that can conflict with reward maximization unless it is explicitly modeled and enforced at a deeper level.
>
> > The faithfulness-as-reward experiments, though conceptually interesting, are underexplored; their instability and optimization dynamics merit deeper quantitative analysis.
>
> Thank you for highlighting the faithfulness-as-reward experiments and agree that the underlying optimization dynamics are an important avenue for deeper analysis. We would like to clarify that, in this section, our use of the term “instability” is not intended to suggest noisy or divergent optimization in the usual sense (e.g., exploding gradients or failure to converge), but rather an instability in the learned strategy with respect to the intended notion of faithfulness. Concretely, even when we enforce a reward that only grants credit when the answer is correct and the chain-of-thought aligns with a “faithful” trace, the policy quickly collapses onto “easy” behaviors (e.g., blindly following right-caption / right-think augmentations) that achieve high reward without actually discriminating between valid and invalid reasoning traces.
>
> > The study relies on a single large-language-model judge (Qwen3-32B) to assess reasoning faithfulness, which may introduce evaluation bias; cross-validation with other judgment models would strengthen the claims.
>
> We agree that having additional evaluators would strengthen the claims and thus we have added faithfulness results across two more judge models, GPT-OSS-120B and Llama-3.1-70B-Instruct. Please see our general comment and updated manuscript in Section 2; overall, we observe that the results are consistent, with very high inter-annotator agreement across judge models.
>
> > While the work discusses potential solutions (e.g., richer reward signals, uncertainty modeling), these remain qualitative suggestions rather than systematically validated interventions.
>
> Please see our response to the related questions below.

---

> ### Author Response · Authors · 2025-11-21
>
> ### Questions
>
> > Have the authors considered developing a formal metric or theoretical framing for “faithfulness drift”? For instance, could the trade-off between accuracy and CoT consistency be modeled via reward attribution entropy or causal influence metrics?
>
> We appreciate the suggestion to formalize “faithfulness drift” and fully agree that a more principled metric or theoretical framing would be valuable. Currently, we track how CoT–answer consistency changes relative to accuracy under different training and augmentation settings—rather than via a formal causal or information-theoretic metric. In light of the reviewer’s suggestion, one concrete quantitative direction we see is to analyze how the model’s final answer probabilities shift with versus without perturbations, conditioned on the same underlying visual input. We would be interested in whether this kind of intervention-based analysis aligns with what the reviewer has in mind, and how it might relate to their notion of “reward attribution entropy.”
>
> > Can the authors provide quantitative evidence—such as variance across seeds, reward gradients, or convergence plots—to substantiate the claim of unstable training dynamics? Could the authors empirically evaluate one or more of the suggested remedies (e.g., uncertainty-aware reward, contrastive consistency loss) to demonstrate their potential effectiveness?
>
> We agree that a more comprehensive study of reward gradients and alternative credit-assignment schemes would be valuable, but given the already large cross-product of models, augmentation settings, and seeds, we were constrained in how many additional RL sweeps we could conduct in this work. Our results instead point to the need to design different algorithms and pretraining pipelines that equip VLMs to use chains of thought as genuinely diagnostic signals, rather than as another channel to steer its output distribution towards a verifiable reward. For example, a promising next step is to first SFT models on explicit reasoning traces that identify and resolve conflicts between visual and textual signals, and only then apply RL to refine robustness. We will emphasize this perspective more clearly in the conclusion, and we view our negative results here as motivation for such future work rather than a fully exhaustive analysis of all possible faithfulness-aware RL designs.
>
> > Have alternative evaluators been tested to confirm that the faithfulness judgments are not artifacts of Qwen3’s inductive biases?
>
> Please see our response in ``Weaknesses’’.
>
> We again thank the reviewer for these thoughtful comments and suggestions, and we look forward to further discussion and feedback. We are also happy to provide additional clarification or targeted analyses if there are specific aspects the reviewer would like us to expand on.

---

### Author Response · Authors · 2025-11-21

Edit (12/03): We’ve once again updated the manuscript with additional results, and appropriately updated our comment below to accurately reflect new figure and table numbers (entropy–based metrics as per the suggestion of Reviewer 2c19). We believe this strengthens the core contributions of our work, demonstrating that unique insights can be gleaned from alternative prompting methods to effectively separate post-trained models based on their training setups. Specifically, these diagnostics reveal how different training setups fundamentally alter the trade-off between internal confidence and responsiveness to external instructions.

We’d like to thank all reviewers for taking the time to provide careful feedback of our work. We have updated our submission with edits and additional results, namely the addition of three new benchmarks and two new evaluator models to validate our approach. The edits are visible in blue font. To summarize, we have made the following changes:

* Section 2: Added analysis on **entropy-based metrics** on all open-source VLMs and benchmarks. Specifically, in **Table 2** we highlight the changes in entropy over the answer tokens when moving from default to stop-think prompting, where models are more uncertain across the board despite varying performance deltas in Table 1.
	* We have also added Appendix D.1 to analyze model behavior through entropy metrics. We find that the "Stop-Think" strategy improves calibration by increasing entropy primarily on incorrect answers, whereas adversarial prompts often induce "confident errors" where models become more certain of wrong predictions. Furthermore, we uncover a trade-off between robustness calibration and faithfulness: "stubborn expert" models show high correlations between base-confidence and robustness (AUROC $>0.95$) but must ignore adversarial chains-of-thought to remain correct, while "obedient reasoners" are highly faithful to the reasoning context but exhibit brittle confidence that fails to predict their vulnerability to perturbations.
* Section 2: Added three new datasets (MMBench, MME-RealWorld-Lite, V*-Bench) which emphasizes more general VQA and real-world understanding rather than primarily spatial relations and addresses concerns raised by Reviewers h5wX and j4PE. The full ablations are given in **Appendix D.2** (Wrong-Think/Wrong-Caption, Stop-Think, Faithfulness and interannotator agreement). We generally see the same trends and provide additional exposition in the pdf.
* Section 2: Added two evaluator models as LLM-as-judges (Llama3.1-70B-Instruct, GPT-OSS-120B) and report interannotator agreement metrics in **Table 3**. We chose these models to span different model families and training data distributions. Reviewers 2c19 and j4PE raised this valid concern about possible evaluation bias introduced by using Qwen3-32B. We put the table below as well, reporting on agreement across judge models (A = GPT-OSS-120B, B = Qwen3-32B, C = Llama3.1-70B-Instruct). Quantities are grouped by dataset and averaged across all open-source VLMs and perturbation settings. Strict 3-way agreement reports the percentage of examples where all three judges agree. Across datasets, we observe consistently high inter-annotator agreement among the three judge models.

| Dataset            | Strict 3-way agree (%) | Cohen's κ(A,B) | κ(A,C) | κ(B,C) | Fleiss' κ |
|--------------------|------------------------|----------------|--------|--------|-----------|
| 3DSRBench          | 92.3                   | 0.880          | 0.846  | 0.823  | 0.850     |
| CVBench            | 94.2                   | 0.899          | 0.883  | 0.870  | 0.884     |
| SpatialMM-Obj      | 93.4                   | 0.889          | 0.868  | 0.865  | 0.874     |
| SpatialMM-Multihop | 94.0                   | 0.876          | 0.860  | 0.823  | 0.853     |
| WhatsUp            | 89.2                   | 0.845          | 0.801  | 0.783  | 0.809     |

* Section 3.1: Expanded and clarified our data augmentation setup for the RL fine-tuning.
* Conclusion: More explicitly discussing why simply incorporating the text perturbations directly in the RL training data and adding a faithfulness-reward doesn’t achieve the desired behavior, and takeaways from this analysis.

We again thank the reviewers for their thoughtful and constructive feedback, which has significantly helped us refine and strengthen this work. We hope that the additional datasets, evaluator models, and ablations clarify our main claims and address the key concerns raised in the reviews. We are also actively working to incorporate additional dataset evaluations for our RL checkpoints (of which there are many more models than we could fully cover in this initial response), but we expect the qualitative trends reported in the current version to hold in these extended experiments as well.

---

### Meta-Review · Area_Chair_bq3G · 2026-01-03

**Summary:**

This paper investigates the robustness and chain-of-thought (CoT) faithfulness of RL-finetuned vision-language models under controlled textual perturbations. Reviewers agreed that the topic is timely and relevant, and that the paper provides a careful empirical diagnosis of how RL fine-tuning can introduce an accuracy-faithfulness tradeoff and increased sensitivity to misleading textual context. However, reviewers expressed mixed opinions on the strength of the contribution. While some found the perturbation framework and empirical findings insightful and practically valuable, others viewed the methodology as largely incremental and primarily diagnostic, lacking theoretical grounding or a principled algorithmic solution. These concerns about novelty, depth of empirical validation, and reuse value ultimately informed my recommendation.

**Reviewer Concerns:**

**Concerns largely addressed by the rebuttal:**

* **Evaluation bias / single judge issue**: The authors added two additional LLM-as-judges and reported inter-annotator agreement, addressing concerns about reliance on a single evaluator.
* **Dataset diversity and task scope**: Additional benchmarks (MMBench, MME-RealWorld-Lite, V\*-Bench) were included, partially alleviating concerns that results were limited to simple spatial reasoning tasks.
* **Clarity of implementation details**: The rebuttal clarified how captions and “initial think” annotations are used during RL fine-tuning.
* **Empirical depth**: Added entropy-based analysis and expanded ablations strengthened the empirical story and improved interpretability.

**Concerns still outstanding:**

* **Limited methodological novelty**: Some reviewers remain unconvinced that the paper goes beyond applying established perturbation and augmentation ideas to a new setting.
* **Primarily diagnostic contribution**: The paper does not propose a reusable training or inference method that demonstrably resolves the identified issues.
* **Lack of theoretical framing**: There is no formal model or principled theory explaining the observed accuracy-faithfulness tradeoff.
* **Robustness gains remain limited**: Even after additional experiments, improvements under augmentation and faithfulness-aware rewards are modest and sometimes unstable.

**Reviewer Scores:**

Reviewer 2c19: Likely remains around 6 (marginally above acceptance threshold); main concerns were partially addressed, but core limitations remain.
Reviewer h5wX: Likely remains 4 (borderline reject); acknowledges improved clarity and added datasets, but still sees limited transfer evidence.
Reviewer j4PE: Raised score from 2 to 4 after rebuttal; the main concern regarding the contribution of the work remains.

---

### Decision · Program_Chairs · 2026-01-26

Reject